# Sex-Specific HLA Alleles Contribute to the Modulation of COVID-19 Severity

**DOI:** 10.3390/ijms252313198

**Published:** 2024-12-08

**Authors:** Serena Spartano, Maria Vittoria Faggiano, Giovanna Guidi, Pino D’Ambrosio, Alessandro Vaisfeld, Agnese Novelli, Salvatore Falqui, Antonella Cingolani, Lorenza Lambertenghi, Alessandro Visentin, Annamaria Azzini, Elda Righi, Enrico Maria Trecarichi, Maria Mazzitelli, Silvano Coletti, Jan Mous, Thomas W. Rademacher, Carlo Torti, Evelina Tacconelli, Massimo Fantoni, Roberto Cauda, Francesco Danilo Tiziano

**Affiliations:** 1Section of Genomic Medicine, Department of Life Sciences and Public Health, Catholic University of Sacred Heart, 00168 Rome, Italy; serenaspartano@gmail.com (S.S.); faggianomariavittoria@gmail.com (M.V.F.); pino.dambrosio01@icatt.it (P.D.); alessandro.vaisfeld2@unibo.it (A.V.); novelliagnese@gmail.com (A.N.); salvatore.falqui@yahoo.it (S.F.); 2Complex Unit of Medical Genetics, Department of Laboratory and Infectivologic Sciences, Policlinico “A. Gemelli” Foundation, 00168 Rome, Italy; 3Section of Infectious Diseases, Department of Safety and Bioethics, Catholic University of Sacred Heart, 00168 Rome, Italy; giovannaguidi10@gmail.com (G.G.); antonella.cingolani@policlinicogemelli.it (A.C.); carlo.torti@unicatt.it (C.T.); massimo.fantoni@unicatt.it (M.F.); roberto.cauda@unicatt.it (R.C.); 4Unit of Infectious Diseases, Department of Medical and Surgical Sciences, Policlinico “A. Gemelli” Foundation, 00168 Rome, Italy; 5Infectious Diseases Division, Department of Diagnostics and Public Health, University of Verona, 37134 Verona, Italy; lorenza.lambertenghi@gmail.com (L.L.); alessandrovisentin2@gmail.com (A.V.); annamaria.azzini@univr.it (A.A.); elda.righi@univr.it (E.R.);; 6Infectious and Tropical Diseases Unit, Department of Medical and Surgical Sciences, University “Magna Graecia”—“R. Dulbecco”, 88100 Catanzaro, Italy; em.trecarichi@unicz.it (E.M.T.); m.mazzitelli@gmail.com (M.M.); 7Chelonia SA, 6900 Lugano, Switzerland; silvano.coletti@chelonia.swiss (S.C.); ian.mous@bluwin.ch (J.M.); 8Division of Infection and Immunity, University College London and Middlesex University, London WC1E 6BT, UK; t.rademacher@ucl.ac.uk; 9Clinical Research Unit, German Center for Infectious Diseases, Tübingen University, 72074 Tübingen, Germany

**Keywords:** SARS-CoV-2, COVID-19, HLA gene variants, next-generation sequencing, genetic association study, biological sex differences, immune system modulation, genetic architecture, case–control study

## Abstract

Severe Acute Respiratory Syndrome Coronavirus 2 (SARS-CoV-2) infection, responsible for Coronavirus Disease 2019 (COVID-19), exhibits a spectrum of clinical manifestations, ranging from asymptomatic to severe pulmonary dysfunction or death. The variability in COVID-19 severity has largely been attributed to the host’s genetic characteristics, suggesting a polygenic genetic architecture, without significant strong evidence of sex-related genetic differences. In this Italian retrospective case–control study, we investigated the association between COVID-19 severity (severe vs. asymptomatic/oligosymptomatic healed individuals) and HLA gene variants, analyzed by next-generation sequencing (NGS). We identified significant HLA alleles (according to the conventional nomenclature), SNPs and haplotypes in the *HLA-B*, *-C*, *-F*, *-DQA1*, *-DRB1*, and *-DRB5* genes associated with COVID-19 severity. Interestingly, these variants showed biological sex-related effects. Also, we identified specific haplotypes associated with COVID-19 severity that are shared by different conventional HLA alleles, indicated here as “super-haplotypes”. These haplotypes had a biological sex-specific impact on disease severity and markedly increased the risk of severe COVID-19 compared to the conventional HLA alleles (odds ratio of up to 15). Our data suggest that the revision of the current HLA nomenclature may help to identify variants with a stronger effect on disease susceptibility and that association studies could benefit from the stratification of patients by biological sex. If replicated in other disease models, these findings could help to define the functional diversity in immune response between sexes, also based on the HLA system. Finally, due to the global pandemic’s mortality rate, we hypothesize here that SARS-CoV-2 may have acted as a natural selection trigger, leading to a drift in HLA allelic frequencies in the general population.

## 1. Introduction

The Severe Acute Respiratory Syndrome Coronavirus 2 (SARS-CoV-2) pandemic has had some prominent characteristics, including extremely high transmissibility, relatively high viral mutability, and variability in individual response to infection [1]. The severity of Coronavirus Disease 2019 (COVID-19) has varied from completely asymptomatic forms to very severe pulmonary dysfunction with respiratory insufficiency or death [2]. The mortality rate of COVID-19 has been very variable in different studies and populations, likely based on viral genotype, standards of care for the clinical assistance of patients, vaccination status and individual characteristics of the host [3]. Among the negative prognostic factors were male biological sex, older age and co-morbidities, as confirmed in a recent meta-analysis [2]. However, co-morbidities correlate more with the risk of death, rather than with disease severity [2].

Regarding the correlation between the genetic background of the host and COVID-19 severity, different case–control genome-wide association studies have been performed. In a recent meta-analysis, 49 SNPs were significantly associated with severe forms of COVID-19 [4].

Some studies have concentrated more on the specific role of the Human Leukocyte Antigen (HLA) system as a severity modulator of COVID-19, identifying some potentially relevant HLA alleles [5]. Some alleles, such as HLA-A02, may reduce the risk of COVID-19, while others, like HLA-C04, could increase severity and mortality [6]. In detail, HLA-C* 04:01 has been identified as a genetic predictor of a severe clinical course of COVID-19, leading to a 2-fold increase in the risk of intubation [7]. In an Italian cohort, carriers of the HLA-B*44 allele had a lower risk of developing severe COVID-19 symptoms [8]; in a Dutch study, the HLA-B*15:03 allele was associated with a reduced risk of infection [3]. Finally, it has been shown that HLA-DRB1*04:01 is more frequent among asymptomatic compared to symptomatic subjects, suggesting a protective role [5]. In a recent review, Hoseinnezhad et al. evaluated the role of HLA genetic variants in COVID-19 susceptibility, severity, and mortality in different populations. These authors suggested that the distribution of HLA alleles in the different populations may account, at least partially, for the diverse outcomes of SARS-CoV-2 infection observed in different geographic areas [6]. To our knowledge, in none of the studies mentioned above were cases and controls stratified according to biological sex.

In the present study, we performed an Italian multicentric case–control study of the association of HLA gene variants with the severity of COVID-19. The study design was prospective–retrospective; subjects healed from COVID-19 were prospectively recruited and retrospectively assigned to case (severe) or control (oligo-asymptomatic) arms. HLA genes (listed in Appendix A) were genotyped by next-generation sequencing (NGS) on gDNA extracted from blood samples collected on Dried Blood Spot (DBS) cards.

## 2. Results

### 2.1. Population Characteristics

We recruited 679 subjects (493 cases), with a case–control ratio of 2.6. The demographic characteristics and inclusion criteria are reported in Table 1. Males represented 67.55% and 42.47% of cases and controls, respectively (*p* < 0.0001). The median age of the cases was 59.24 years, and that of the controls was 52.55 (*p* < 0.0001).

### 2.2. HLA-DRB1 Is the Most Polymorphic Gene

To assess the intrinsic genetic variability of *HLA* genes in our population, we related number of identified variants and size of the sequenced region per each gene. As shown in Figure 1, *HLA-DRB1* displayed the highest number of variants/sequenced base pairs (bp). *HLA-A*, *-B*, and *-C* displayed a similar polymorphic rate; *HLA-E*, *-F*, and *-G* appeared to be the least variable (ranging between 0.018 and 0.043 variants/bp); most class II genes displayed a similar range of variation.

### 2.3. HLA-A, -B, -C, -DPA1, -DPB1, -DQB1, -DRB1 Alleles Are Associated with COVID-19 Severity in a Biological Sex-Specific Manner

For conventional HLA genotyping and nomenclature (HLA-alleles, ImMunoGeneTics project, IMGT), Ref. [9] data were analyzable for 670 individuals. Results are reported in Table 2 and detailed in Appendix A. The alleles indicated below in *HLA-A*, *-B*, *-C -DPA1*, *-DPB1*, *-DQB1* and *-DRB1* were associated with COVID-19 severity.

The stratification of the cohorts by biological sex led to the identification of HLA alleles with a sex-specific effect; the same allele might be indifferent, protective or of risk, in one or the other sex (Table 2), or even non-significant in the whole population (see HLA-C*12 or *15 as examples in Table 2).

### 2.4. Single HLA-SNPs Are Associated with COVID-19 Severity

We identified 5578 Single-Nucleotide Polymorphisms at the HLA locus (HLA-SNPs), 118 significantly associated with COVID-19 severity (FDR ≤ 0.05). Among the significant variants, we prioritized only those in the coding regions and the intronic predicted to affect exon splicing, irrespective of the allelic frequency. We identified 28 variants that were confirmed by Sanger Sequencing (see Table 3, Appendix A). Among the class I genes, *HLA-B*, *-C*, and *-F* genes were significantly associated with the severe disease phenotype, and among the class II, *HLA-DQA1*, *-DRB1*, and *-DRB5*. The significant HLA-SNPs, their relative odds ratio (OR), 95% confidence intervals (CI) and *p*-values are reported in Table 3. The OR values for the single variants ranged from 1.02 to 3.62 for risk SNPs, and from 0.42 to 0.77 for the protective.

### 2.5. Age and Biological Sex Impact on the Significance of HLA-SNPs on COVID-19

Due to the significant difference between cases and controls in age and biological sex, we included these two variables as co-variates in a multinomial logistic regression model; this analysis led to a reduction in the number of significant HLA-SNPs. The remaining are indicated in bold in Table 3. According to this model, *HLA-DQA1* variants lost significance.

We analyzed the single or combined effect of each HLA-SNP by comparing allelic and genotypic frequencies, respectively, by chi-squared test. This approach was used to evaluate whether the COVID-19 severity risk was influenced by the homozygous or heterozygous state for each variant. To assess the possible isoallelic influence of the single HLA-SNPs (i.e., if SNPs present in the two alleles might have an additive effect on COVID-19 severity risk), we used a log-additive model. Both analyses were conducted in the whole population of cases or controls, or stratified by biological sex. We finally compared the allelic frequency of each HLA-SNP in males and females, independent if cases or controls (column labeled as M vs. F in Table 4).

From the analysis of the two cohorts, *HLA-B*, *-C*, *-DRB1* and *-DRB5* were significantly associated with COVID-19 severity, whereas *HLA-F* and *-DQB1* lost significance. In most cases (except for few variants), the COVID-19 risk was not influenced by the heterozygous or homozygous state, and we did not find consistent evidence of isoallelic influence for the single SNPs, as suggested by the OR values.

The stratification of the cohorts by biological sex led to two relevant observations:

(1) Risk genes were different in males and females (Table 4).

As an example, *HLA-C* was associated with the phenotype in males only, *HLA-B* in females.

(2) Allelic and genotypic frequencies of HLA-SNPs were significantly different in the two sexes when stratified by cases and controls but not in the whole study population, as expected for autosomal loci, ruling out possible bias in patients’ enrollment (Table 4).

### 2.6. “Super-Haplotypes” Are Shared Among Different HLA Alleles

To evaluate the potential combinatorial *cis* effect of the significant HLA-SNPs, i.e., whether variants located in the same allele might reciprocally interact, we inferred haplotypes and gametic phases per each HLA gene [10]. Haplotypes were arbitrarily numbered from 1 (haploptype ID #), and are listed in Appendix A. We identified some haplotypes that were associated with COVID-19 severity (listed in Table 5) in a biological sex-specific manner; the risk haplotypes were different in males or females, or might display opposite behavior in biological sexes.

As examples, *HLA-B* haplotype#3 and #5 were protective in males, of risk in females, non-significant in the whole population; *HLA-C* haplotype#9 and #17 were significant in females only (risk haplotypes). Also, in these analyses, the haplotypic frequencies in the whole population were not significantly different between males and females, ruling out possible sample biases.

To identify the conventional HLA alleles bearing each significant haplotype, we aligned the nucleotidic positions characterizing each haplotype with the corresponding positions in the HLA cDNA sequences contained in the HLA-IMGT dictionary. Surprisingly, we found that:(1)the same haplotype was present in different HLA-alleles;(2)only a small number of haplotypes were found in the dictionary (see Appendix A).

Following these observations, we propose here to define “super-haplotypes” (HLA-SH), those shared by different HLA alleles.

## 3. Discussion

Most variability of COVID-19 has been attributed to the genetic characteristics of the host, although the association studies performed so far have not revealed major genetic loci or variants with strong modulating effect, suggesting a polygenic genetic architecture; additionally, there is no clear evidence of biological sex-related differences, except for very rare variants, located in the X-chromosome [11].

To evaluate the specific role of HLA genes in the modulation of COVID-19 severity, we have conceived the present study as retrospective with prospective recruitment of patients healed from COVID-19, with the aim to clearly identify dichotomic groups (a-/oligosymptomatic vs. severe subjects) and maximize the chance of detecting relevant predisposing variants in the HLA locus. On the other hand, this strategy prevented us from studying the extreme side of the spectrum of the disease, the patients who died of COVID-19.

Regarding the choice of the experimental technology, we opted for the NGS of the exons and the flanking intronic regions of HLA genes (excluding known pseudogenes); this approach provided the unquestionable advantages of reducing costs/sample. On the other hand, the huge increase in the number of variants identified, including the intronic, has enormously increased the complexity of result interpretation; it is conceivable that most SNPs are irrelevant to the phenotype or are simply in linkage disequilibrium with others that are functionally relevant. In the present study, to facilitate data interpretation, we focused on variants located in the coding regions only, although some intronic variants were significantly associated with COVID-19 severity. This approach might have led to the loss of variants with a splicing-modulating effect; it is well known that alternative splicing is a common event in the HLA genes, although the biological meaning and the regulation of the inclusion/exclusion of specific exons are largely unknown.

Our data have revealed an unexpected variable degree of polymorphism of each HLA gene; the evolutionary reason for a lesser extent of variability of *HLA-E*, *-F*, and *-G*, up to 8-fold lower than the other class I genes, is unknown (see Figure 1). These characteristics are even more evident in the class II genes, with some genes being extremely variable, more so than *HLA-A*, *-B* and *-C*. We wonder whether this could be related to intrinsic characteristics of the HLA locus or could rather be attributed to a negative selection on the variation of some genes, related to the specific function in the immune system.

Regarding the association with disease severity, we first identified significantly associated HLA alleles and found those reported in Table 2. Among the class I genes, *HLA-A*, *-B*, and *-C* were associated with COVID-19 severity, among the class II genes, *HLA-DPA1*, *-DPB1*, *-DQB1*, *-DRB1*. The significant alleles conferred a moderate increase/decrease in the OR of severe forms, up to 2-fold. None of these alleles was among the previously identified ones, reported in the literature; this discrepancy could be related to different inclusion criteria, technical differences in genotyping or different allelic frequencies in the studied populations. Differently from previously published studies, we evaluated the biological sex-specific effects of HLA alleles, which led to a consistent improvement in patient risk assessment (Table 2).

Subsequently, we analyzed our dataset using a classical genetic case–control association model. This approach led to the identification of significant HLA-SNPs and to consistently different results than the previous approach; the association of some genes lost significance (*HLA-A* and all class II genes except *-DRB5*), whereas other genes became significantly associated with COVID-19, namely *HLA-F*, *-DQA1*, and *-DRB5*. Biological sex stratification led to even more striking results; risk genes differed in males and females according to a quite clear dichotomous model (Table 4).

To address the role of the *cis* effect of the significant variants in each gene, we first inferred the gametic phase of these variants, and identified the HLA-SH, as indicated in the results. Importantly, these haplotypes have an opposite effect in the two sexes. Only a few had a biological sex-independent impact. The identification of HLA-SH markedly improved the risk assessment compared to the conventional genotyping, since the differential risk of severe forms of COVID-19 was increased up to 5-fold between males and females, as in the case of HLA-SH#1 of the *HLA-DQA1* gene that is of high risk in females.

Besides the translational impact of our findings, in terms of individual risk assessment for infected subjects, which may still be useful during the current epidemic phase of COVID-19, there are several lessons we have learned from the present study. First, the identification of HLA-SH strongly suggests the need to revise the standard nomenclature of HLA alleles and the technical approaches to HLA genotyping, through validation in exploratory scenarios.

Our study has also revealed the limitations of the gDNA-based approach in terms of interpretation of the HLA variability in the general population, of the combinatorial effect of *cis*-variants in the same HLA gene and of possible isoallelic influence of the homologous gene in *trans*. The most used NGS platforms are based on short read sequencing that do not allow the definition of the gametic phase of the single variants. The spreading of long-read sequencing of gDNA may help address this point and define the combinatorial effect of the different HLA gene variants, in *cis* on the same chromosome. On the other hand, this approach will be unlikely to help define the role of intronic variants, namely in the alternative splicing modulation, which may play a key role in the immune system modulation. As an example, it has been shown that the skipping of exon 4 of *HLA-G* leads to the production of a soluble secreted isoform that apparently does not have an antigen presentation function but instead has an immunomodulatory effect, via the regulation of Natural Killer (NK) cells [12]. This phenomenon is less known in other HLA genes but, in principle, a variable splicing pattern can produce isoforms with different functions than those classically attributed to the MHC molecules [13]. We propose implementing, at least in an exploratory phase, HLA genotyping on cDNA from whole blood. To our knowledge, there are few such studies reported in the literature [14,15,16]. Even though this approach does not allow the identification of intronic variants, it provides experimental data on the gametic phase of the coding SNPs and on the qualification and quantification of the alternative mRNA isoform repertoire in vivo.

From the immunology point of view, we provide here evidence of biological sex-specific differences in disease susceptibility related to HLA genes in humans. It is well known that the immune response is different in males and females, and that there are clear differences in the epidemiology of some infectious and autoimmune diseases [17]. Some of the differences in immune response have been very recently attributed to the effect of sex hormones, namely androgens [18]. However, we could not find in the literature studies including the differential analysis of HLA genotype according to biological sex. Similarly, we could not find raw data of published association studies on autoimmune or other HLA-related diseases, including sex-specific allelic frequencies, to perform meta-analyses. It could be of extreme interest to assess whether our findings are replicated in other disease models, beyond COVID-19, since this could shed some additional hints on the functional diversity in the immune system function between males and females.

Our study identified a specific HLA-SNP and haplotype of the non-classical class I gene *HLA-F*, associated with disease severity. More information is needed on the role of this HLA gene in human disease, even though relatively recent, preliminary data suggest the involvement in viral infection [19]. To our knowledge, *HLA-F* has been mainly related to the embryo tolerance [20] and is involved in immune system regulation through the modulation of NK response [19]. In our population, *HLA-F* is among the least variable genes; a single variant, having an allelic frequency of 0.04 in the global population and 0.06 in female cases, confers a 5-fold increased risk of severe COVID-19. The functional role of this variant and, more in general, of *HLA-F* in disease will require future additional dedicated studies.

As a final remark, from the point of view of the classical genetics of populations, our study has provided novel potentially straightforward insights. As expected, when analyzing our population as a whole (without the stratification by cases or controls), allelic and genotypic frequencies of HLA alleles, HLA-SNPs and HLA-SH are not different between males and females, as expected for autosomal loci in the absence of selective pressure. When stratifying by cases and controls, we observed a statistically significant difference in frequencies between males and females; we can hypothesize an “acute” biological sex-specific natural selection event based on HLA genotype. Even if we could not analyze samples from individuals who died of COVID-19, we can hypothesize the existence of HLA “deadly” alleles that were lost in the general population. If this hypothesis is correct, the huge number of COVID-19 victims worldwide (in the order of several million, underestimated due to the lack of punctual data from many countries) may have led to a consistent drift in the HLA allelic frequencies in the general population. A similar phenomenon may have occurred during the Spanish flu pandemics of 1918–1920, with an estimated number of deaths of 50M people worldwide out of an estimated population of 1.8 billion persons [21].

Among the differences between the two pandemics is that most victims of the Spanish flu were subjects of reproductive age (18–35 years) [21], thus likely changing allelic frequencies by a consistent extent of phenotypically relevant genes over the next generations. In the case of COVID-19, most victims were over reproductive age.

We are well aware that the monogenic prediction model of COVID-19 severity is an over-simplification. However, we wonder whether COVID-19 may have led to the decanalization (i.e., to the perturbation of the gene–environment homeostasis due to large effect-size environmental changes or genetic variants) [22] of the immune response to some environmental triggers, mediated by the HLA locus. A similar mechanism has been evoked to explain the increase in type II diabetes in Greenlandic populations, in response to an increase in carbohydrate intake with diet and the presence of a single *TBC1D4* SNP (rs61736969) [23]. To address this point, due to the key role of HLA genes in modulating the susceptibility to most diseases, it will be advisable to monitor possible changes in the epidemiology of immune-mediated conditions over the next few years.

Our study presents some limitations. During the enrollment phase, the epidemiology of the pandemics changed consistently, due to the drift of SARS-CoV-2 viral variants. During the first part of the enrollment, patients may have been mainly infected by the wild-type and alpha strains, while during 2022, the main circulating variant was the BA (Omicron variant) [24]. Due to the healed status of the patients, we could not genotype viral variants in our cohorts; viral variant assessment was not routinely performed at the time of the infection. Additionally, starting from November 2021, the epidemiology of COVID-19 changed when vaccines became available, leading to a marked slowdown in recruitment rate.

In conclusion, our study provides some hints that will likely open new perspectives for further studies in understanding HLA-mediated immune modulation. Starting from this evidence, new scenarios could be opened in understanding HLA-mediated immune modulation in response to environmental HLA effect among biological sexes, beyond the SARS-CoV2 pandemics.

The identification of individual genetic fingerprints, predictive of the severity of disease in the case of exposure to pathogens, might contribute, during interpandemic times, to provide operational procedures for the early therapeutic management protocols and vaccination campaigns.

## 4. Materials and Methods

### 4.1. Study Design

We conducted a retrospective case–control association study aimed at investigating the role of HLA variants in COVID-19 severity. Healed subjects were retrospectively assigned to case (severe) or control (oligo-/asymptomatics) arms and prospectively enrolled between 7 December 2020 and 12 May 2022.

### 4.2. Recruiting Centers

The study was conducted in 3 Italian University Hospitals, located in Rome, Verona and Catanzaro, and was approved by the ethics committees of the three institutions. Cases were recruited among healed patients (subjects who tested positive for the presence of SARS-CoV-2 genome and subsequently tested negative in at least two repeated samplings), in follow-up at the post-COVID outpatients’ units; controls were recruited among the oligo-/asymptomatic relatives.

The enrollment criteria were as follows:

Cases: Hospitalized patients who displayed at least one of the following clinical findings:Oxygen saturation (SaO2) < 94% on room air; <90% in presence of known chronic hypoxic conditions or receiving chronic supplemental oxygen;Respiratory rate > 24 breaths/min.

Controls: Subjects who did not require hospitalization and who displayed few or no signs/symptoms associated with COVID-19, without a relevant impairment in clinical conditions or quality of life.

Sample size and study power were calculated by Genetic Power Calculator [25,26]. We have conservatively considered a minimum allele frequency (MAF) of 5% for each genetic variant in the population and a prevalence of the severe phenotype of about 10%. Assuming a population study of 500 subjects/arm and ratio 1:1 cases:controls, the study would have achieved a power of 80% (at an α value of 0.05) for odds ratios > 2. The expected prevalence of Acute Respiratory Distress Syndrome (ARDS, 14–15% of infected subjects) and the choice of dichotomous groups (ARDS vs. a-/oligosymptomatic) were expected to provide an increase in the study power.

Study variables were age, biological sex, and severity of COVID-19 symptoms.

### 4.3. Molecular Methods

Blood samples were collected on Dried Blood Spots (DBS) from finger-sticking. We opted for this solution in order to simplify blood sample collection, storage and delivery from the recruiting centers.

gDNA extraction from DBS was performed as previously reported [27], using three 3.5 mm punches per sample and the DNA Blood Spot LH kit (Perkin Elmer, Shelton, CT, USA).

NGS barcoded libraries were prepared using the AmpliSeq approach. Briefly, we designed an AmpliSeq panel (https://www.ampliseq.com, accessed on 1 May 2020) for the HLA genes listed in Appendix A along with their relative accession numbers (IAD_199394_182), including 819 amplicons. The coverage of the coding region of each gene was 100%, plus we included at least 50 bp of the intronic region flanking each exon.

For library preparation and purification, we used the Ion AmpliSeq^TM^ Library Kit plus (ThermoFisher Scientific, Waltham, MA, USA); for barcoding, we used the IonCode Plates (ThermoFisher Scientific, Waltham, MA, USA), 96 barcoded libraries/run. For purification, we used the AMPure XP Beads (Beckman Coulter, Brea, CA, USA), according to the AmpliSeq protocol instructions. Library preparation was performed in a semi-automatized pipeline, using a home-developed script for the Sciclone NGS Instrument (Perkin Elmer, Shelton, CT, USA).

Following purification, for library concentration balance and pool preparation, we used the Ion Library Equalizer kit (Thermofisher, Waltham, MA, USA), according to the manufacturer’s protocol. The ION Chef Instrument (ThermoFisher Scientific, Waltham, MA, USA) was used for emulsion PCR preparation and chip loading, using the ION Ion PI Hi-Q Chef Kit (ThermoFisher Scientific, Waltham, MA, USA). Sequencing was performed by the ION Proton Instrument, according to the manufacturer’s protocols. Ninety-six samples were run per chip. Basecalling, alignment to GRCh37/hg19 genome, and generation of the variant caller file (VCF) were performed by the resident Torrent Suite version 5.6. Variants (HLA-SNPs) were annotated by the ION-Reporter suite (version 5.8, https://ionreporter.thermofisher.com, accessed on 1 January 2022). We find a total number of 5578 different SNPs (see Appendix A) in our cohort, using the default Ion-Reporter parameters.

### 4.4. NGS Metrics

Quality metrics for NGS data were the following: per base Q30 and average read length between 125 and 275 bp (based on panel characteristics). Only samples with average depth ≥ 30× and uniformity ≥ 80% were analyzed. Samples that did not meet these quality standards were re-run; multiple BAM files of the same sample were merged using Samtools (version 1.13) [28].

The average number of reads per sample was 401,781 ± 295,638, the average depth 4921 ± 2942×, the mean uniformity, 82.51% ± 26.47%.

### 4.5. Sanger Sequencing

Significant variants, located in the coding region or predicted to affect exon splicing, were confirmed by Sanger sequencing.

Briefly, we designed the specific primer pairs, flanking the specific SNP/SNPs of interest, reported in Appendix A. For all PCR reactions, we used the 2× PCR Taq^®^ DNA Polymerase (Promega Co., Madison, WI, USA), 10 µM of each primer and 30 ng of gDNA, in a final volume of 12 µL. Thermal cycling conditions were as follows: 95 °C, 5′; 35× (95 °C, 45″; variable annealing temperature indicated in Appendix A, 45″; 72 °C, 30″); 72 °C, 5′. PCR products were analyzed by agarose gel (1.5% in TAE 1×) standard electrophoresis, and stained with ethidium bromide.

PCR products were purified by exonuclease and alkaline phosphatase (ExoSAP-IT, ThermoFisher Scientific, Waltham, MA, USA) before sequencing, by incubation at 37 °C, 15′ and enzyme heat inactivation (80 °C, 15′). Sequencing reactions were performed by the BigDye Terminator v3.1 Cycle Sequencing Kit (ThermoFisher Scientific, Waltham, MA, USA), in a final volume of 10 µL, according to the manufacturer’s protocol. Sequencing reaction products were purified using the BigDye XTerminator Purification Kit (ThermoFisher Scientific, Waltham, MA, USA), according to the manufacturer’s protocol. Capillary electrophoresis was performed on the Applied Biosystems 3130 Genetic Analyzer, and analyzed by the sequencing analysis software (Version 7.1, ThermoFisher Scientific, Waltham, MA, USA). The samples were electrophoresed, and the resulting data were analyzed to obtain sequence reads (Appendix A).

### 4.6. Bioinformatics and Statistical Analysis

For canonical HLA-genotyping, we used HLA-VBSeq and the IMGT/HLA database21 (release3.15.0) [9].

An in-house Python3.0 script was used to process in batch HLA-VBSeq input files, to write relative output files and to perform statistical analysis of canonical genotyping results.

The association analysis was based on generalized linear models and performed using R4.1.2 (https://cran.r-project.org, accessed on 1 April 2022), a framework and coding language widely used for data analyses. We specifically made use of SNPassoc (https://cran.r-project.org/web/packages/SNPassoc/index.html, accessed on 1 April 2022), an R package (i.e., a collection of functions and other R objects) comprising multiple tools for genetic association studies. For each SNP, single alleles were set as independent variables; we computed the OR associated to the mere presence of the allele, irrespective of the dosage (heterozygous or homozygous) on the basis of the allelic frequency, by chi squared testing on contingency tables. We then assessed for the possible incremental effect of zero, one or two copies of the SNP (i.e., wild-type homozygous, heterozygous, and variant homozygous) in a log-additive model.

Subsequently, we compiled a logistic regression model in IBM SPSS 21.0.0.0 (https://www.ibm.com/products/spss-statistics, accessed on 1 April 2022) setting the genotypes at each HLA gene, age and biological sex as co-variates, to evaluate any combined effect given by age and/or sex. We used a false-discovery-rate (FDR) control for multiple testing adjustment (FDR < 0.05) after calculating nominal *p*-values for each locus, as customary for genetic association studies. In brief, FDR is the expected proportion of “discoveries” (rejected null hypotheses) that are false (i.e., incorrect rejections of the null).

Haplotypes were predicted using statistically significant SNPs as inputs into Arlequin v3.0, designed for inferring gametic phase starting from NGS output data [24]. Association analysis between haplotypes and severity was performed by chi squared testing on haplotypic frequencies.

Sex-specific statistical analyses were performed by means of chi-squared testing on allelic frequencies contingency tables for each HLA allele, SNP and haplotype. In addition, chi-squared testing was also performed for each allele on contingency tables built by dividing the whole population by sex rather than phenotype, in order to assess possible differences in allele distribution between sexes. All table operations subsequent to the first R analysis were performed using Python3.0 (https://www.python.org/download/releases/3.0/, accessed on 1 April 2022) custom scripts, making use of the “CSV” (Comma-Separated Values, Python built-in), and “SciPy” (https://scipy.org/, accessed on 1 April 2022) libraries. These analyses include computation of OR and CI for HLA alleles, SNPs and haplotypes, in the whole population and in the sex-stratified ones.

A Python3.0 script was also created to compare HLA-SH and .fasta sequences of conventional HLA-alleles, available on the IMGT website (https://www.ebi.ac.uk/ipd/imgt/hla/, accessed on 1 September 2022), to identify correspondences between HLA-SH and HLA alleles.

## Figures and Tables

**Figure 1 ijms-25-13198-f001:**
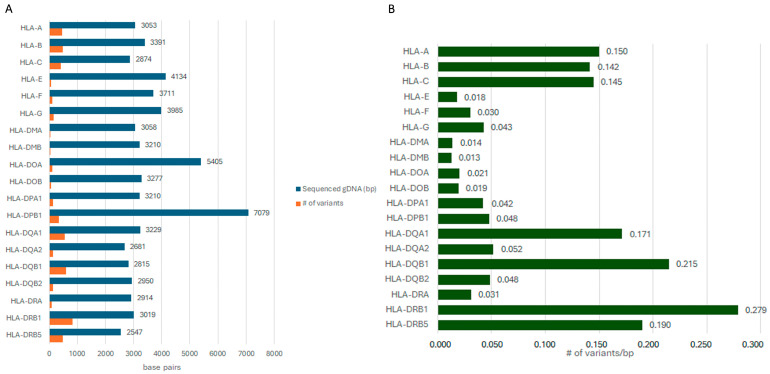
(**A**): Size of the coding region of each gene (bp) vs. # of variants. (**B**): Ratio between # of variants and amount of sequenced gDNA.

**Table 1 ijms-25-13198-t001:** Inclusion criteria for cases/controls and sample distribution.

	Inclusion Criteria	M # (%)	F # (%)	Total # (%)	Median Age Years (±IQR)
**Cases**	Oxygen saturation: <94% on room air <90% with known chronic hypoxic conditions or Respiratory rate > 24/min	333 (67.55)	160 (32.45)	493 (73)	59.24 (11.45)
**Controls**	Few or no signs/symptoms associated with COVID-19 or No relevant impairment in quality of life	86 (42.47)	100 (57.53)	186 (27)	52.55 (16.73)

M: males; F: females; IQR: interquartile range.

**Table 2 ijms-25-13198-t002:** HLA alleles significantly associated with COVID-19 severity. The two cohorts were also stratified according to the biological sex.

Gene	Allele #	Whole Population						Males						Females						M vs. F
		# of Alleles		OR	CI		*p*	# of Alleles		OR	CI		*p*	# of Alleles		OR	CI		*p*	*p*
		Cases (*n =* 970)	Ctrls (*n =* 284)		Low	Up		Cases (*n =* 654)	Ctrls (*n =* 172)		Low	Up		Cases (*n =* 316)	Ctrls (*n =* 198)		Low	Up		
*A*	*24	**151**	**42**	**1.44**	**1**	**2.07**	**0.05**	**104**	**15**	**1.98**	**3.5**	**1.12**	**0.02**	47	27	1.11	1.84	0.66	0.8	1.00
*B*	*14	**35**	**26**	**0.5**	**0.29**	**0.83**	**0.01**	**19**	**16**	**0.29**	**0.58**	**0.15**	**5*10^-4^**	16	10	1	2.26	0.66	1	0.48
*B*	*55	**17**	**14**	**0.45**	**0.22**	**0.93**	**0.03**	10	6	0.43	1.2	0.15	0.18	7	8	0.54	1.51	0.19	0.35	0.25
*C*	*08	**22**	**17**	**0.47**	**0.25**	**0.9**	**0.02**	**11**	**12**	**0.23**	**0.53**	**0.1**	**5*10^-4^**	11	5	1.39	4.07	0.66	0.73	0.73
*C*	*12	120	51	0.87	0.61	1.23	0.43	92	21	1.18	1.95	0.71	0.61	**28**	**30**	**0.54**	**0.94**	**0.66**	**0.04**	0.20
*C*	*15	15	11	0.5	0.23	1.11	0.08	**6**	**6**	**0.26**	**0.8**	**0.08**	**0.03**	9	5	1.13	3.43	0.66	1	0.10
*DPA1*	*01	**636**	**221**	**1.28**	**1**	**1.64**	**0.05**	423	108	1.09	1.54	0.77	0.71	**213**	**113**	**1.56**	**2.24**	**0.66**	**0.02**	0.75
*DPA1*	*02	318	141	0.79	0.62	1.02	0.07	219	60	0.94	1.34	0.66	0.8	**99**	**81**	**0.66**	**0.95**	**0.66**	**0.03**	0.64
*DPB1*	*17	23	13	0.67	0.33	1.33	0.25	18	3	1.59	5.48	0.46	0.63	**5**	**10**	**0.3**	**0.9**	**0.66**	**0.05**	0.68
*DPB1*	*416	4	4	0.38	0.09	1.52	0.16	**1**	**3**	**0.09**	**0.83**	**0.01**	**0.04**	3	1	1.89	18.28	0.66	0.97	0.5
*DQB1*	*05	**229**	**108**	**0.75**	**0.57**	**0.98**	**0.04**	148	51	0.69	1.01	0.48	0.07	81	57	0.85	1.27	0.57	0.49	0.26
*DRB1*	*10	21	11	1.91	0.22	16.42	0.55	**11**	**8**	**0.35**	**0.89**	**0.14**	**0.04**	10	3	2.12	7.82	0.66	0.38	0.79

HLA alleles in our populations were represented as absolute number (#) and percentage (%). No statistically significant differences were seen in HLA allele distribution between sexes in the whole population, ruling out possible sampling bias (see the *p*-value in M vs. F column). Different groups were compared by χ^2^ test. The protective/risk effect of each HLA allele was evaluated by the odds ratio (OR) and lower (Low) and upper (Up) confidence interval (CI). Differences were considered statistically significant for *p <* 0.05. The significant HLA alleles, in at least one category, are indicated in bold in coloured cells.

**Table 3 ijms-25-13198-t003:** HLA-SNPs significantly associated with COVID-19 severity in chi squared testing.

	*Gene*	gDNA (chr6:)	dbSNP ID	cDNA	a.a. Change	Exon	Effect	OR	CI		*p*
									Low	Up	
**Class I**											
	*HLA-B*	31324086G>C	rs709054	c.477C>G	p.Ala159=	3	R	1.02	0.47	2.23	0.04
		31324145A>C	rs9266150	c.418T>G	p.Tyr140Asp	4	R	1.86	1.11	3.11	0.01
		31324523T>C	rs1131214	c.285A>G	p.Ala95=	2	P	0.44	0.24	0.82	0.01
		31324526CTG>GGA	n.a.	c.280CAG>TCC	p.Gln94Ser	2	P	0.48	0.27	0.85	0.01
		31324535GTA>CAT	n.a	c.271TAC>ATG	p.Tyr91Met	2	P	0.5	0.29	0.88	0.02
		31324539A>T	rs1131202	c.269T>A	p.Ile90Thr	2	P	0.44	0.24	0.82	0.01
		**31324542T>C**	**rs1131201**	**c.266A>G**	**p.Gln89Arg**	**2**	**P**	**0.42**	**0.23**	**0.77**	**0.01**
	*HLA-C*	31237124T>C	rs1130838	c.1087T>C	p.Thr363Ser	7	R	3.62	1.05	1.77	0.02
		31237162C>G	rs35708511	c.1049C>G	p.Cys350Ser	7	R	1.36	1.03	1.76	0.03
		31237773T>C	rs1130947	c.985T>C	p.Thr329Ala	5	R	1.85	1.44	2.37	<10^−4^
		31237774T>G	rs41540512	c.984T>G	p.Val328=	5	R	1.85	1.44	2.37	<10^−4^
		31237779C>T	rs146911342	c.979C>T	p.Val327Leu	5	R	1.85	1.52	1.1	0.02
		31237858T>C	rs34794906	c.900T>C	p.Pro300=	5	R	1.41	1.06	1.9	0.02
		31237987C>T	rs41556321	c.895C>T	p.Glu299Ter	4	R	1.48	1.05	2.09	0.02
		31238009TT>CG	rs1131014	c.872TT>CG	p.Gln291Pro	4	R	1.31	1.03	1.66	0.02
		31238027C>A	rs41540117	c.855C>A	p.Met285Ile	4	R	1.47	1.06	2.04	0.01
		31238029T>C	rs2308622	c.853T>C	p.Met285Leu	4	R	1.28	1.01	1.62	0.04
		31238138C>T	rs1050320	c.744C>T	p.Gln248His	4	R	1.34	1.01	1.77	0.03
		**31239449C>G**	**rs28626310**	**c.270C>G**	**p.Lys90Asn**	**2**	**P**	**0.62**	**0.47**	**0.82**	**<10^−4^**
		31239501G>T	rs1050409	c.218G>T	p.Ala73Glu	2	R	1.34	1.00	1.81	0.05
	*HLA-F*	**29694777G>A**	**rs17875384**	**c.1153G>A**	**p.Arg385Gln**	**7**	**R**	**3.62**	**1.28**	**10.25**	**0.01**
**Class II**											
	*HLA-DQA1*	32609806C>T	rs707952	c.388C>T	p.Thr130Ile	3	P	0.72	0.52	0.99	0.04
		32609952T>C	rs707949	c.534T>C	p.Phe179Leu	3	P	0.69	0.49	0.96	0.02
		32609969T>G	rs707963	c.551T>G	p.Asp184Glu	3	P	0.68	0.49	0.95	0.02
		32609974T>G	rs707962	c.556T>G	p.Ile186Ser	3	P	0.66	0.47	0.92	0.01
	*HLA-DRB1*	32551939GG>AT	rs16822512	c.317CC>AT	p. Thr106Asn	2	P	0.77	0.61	0.97	0.02
		**32552092A>T**	**rs1059569**	**c.174T>A**	**p.Phe55Tyr**	**2**	**R**	**1.52**	**1.52**	**1.08**	**0.03**
	*HLA-DRB5*	**32487256G>A**	**rs112401921**	**c.543C>T**	**p.Asp181=**	**3**	**P**	**0.48**	**0.25**	**0.94**	**0.03**

OR, lower (Low) and upper (Up) CI and *p*-values are indicated. Bolded SNPs in coloured cells were significant also in a multinomial logistic regression model, including age and biological sex as co-variates. gDNA: precise genomic position according to the hg19 version of the reference genome assembly (https://www.ncbi.nlm.nih.gov/datasets/genome/GCF_000001405.13/ accessed on 1 July 2022); dbSNP ID: SNP reference number as of dbSNP database (https://www.ncbi.nlm.nih.gov/snp/ accessed on 1^st^ July 2022); R: risk allele; P: protective allele

**Table 4 ijms-25-13198-t004:** Results of HLA-SNP multivariate analysis. The two cohorts were also stratified according to the biological sex.

Gene	gDNA Coordinates (dbSNP rsID)	Log Additive Model				Genotypic Frequencies												Allelic Frequencies												M vs. F
						Global				Males				Females				Global				Males				Females				
		OR	Low	Up	p	OR	Low	Up	p	OR	Low	Up	p	OR	Low	Up	P	OR	Low	Up	p	OR	Low	Up	p	OR	Lower	Upper	p	p
*HLA-B*	31324086G>C (rs709054)	**1**	**0.5**	**2.23**	**0.05**	1.09	0.78	1.52	0.67	1.11	0.69	1.77	0.14	1.15	0.7	1.88	0.61	1.12	0.87	1.46	0.39	1.63	0.76	1.53	0.71	1.1	7.41	1.63	0.68	0.23
	31324145A>C (rs9266150)	**1.9**	**1.1**	**3.11**	**0.01**	**1.8**	**1**	**3.1**	**0.04**	**2.2**	**1**	**4.8**	**0.05**	1.5	0.59	3.82	0.48	**2**	**1.6**	**3.41**	**0.01**	1.84	0.93	3.67	0.10	1.68	0.68	4.13	0.29	0.07
	31324523T>C (rs1131214)	**0.4**	**0.2**	**0.82**	**0.01**	**0.5**	**0.2**	**0.9**	**0.02**	0.62	0.26	1.47	0.22	**0.2**	**0.1**	**0.73**	**0.01**	**0.4**	**0.2**	**0.8**	**0.01**	0.56	0.25	1.26	0.16	**0.2**	**0.1**	**0.75**	**0.01**	0.73
	31324526CTG>GGA (n.a.)	**0.5**	**0.3**	**0.85**	**0.02**	0.79	0.51	1.23	0.29	0.67	0.3	1.52	0.36	**0.3**	**0.1**	**0.7**	**0.01**	**0.5**	**0.3**	**0.85**	**0.02**	0.61	0.29	1.32	0.19	**0.3**	**0.1**	**0.8**	**0.01**	0.88
	31324535GTA>CAT (n.a)	**0.5**	**0.3**	**0.88**	**0.03**	**0.5**	**0.3**	**0.9**	**0.03**	0.71	0.32	1.58	0.38	**0.2**	**0.1**	**0.78**	**0.01**	**0.5**	**0.3**	**0.88**	**0.02**	0.63	0.3	1.37	0.28	**0.3**	**0.1**	**0.8**	**0.01**	0.8
	31324539A>T (rs1131202)	**0.4**	**0.2**	**0.82**	**0.01**	**0.5**	**0.2**	**0.9**	**0.02**	0.69	0.26	1.47	0.32	**0.2**	**0.7**	**0.73**	**0.01**	**0.4**	**0.2**	**0.81**	**0.01**	0.56	0.25	1.26	0.16	**0.2**	**0.1**	**0.75**	**0.01**	0.73
	31324542T>C (rs1131201)	**0.4**	**0.2**	**0.77**	**0.01**	**0.4**	**0.2**	**0.8**	**0.01**	0.54	0.24	1.25	0.15	**0.2**	**0.7**	**0.73**	**0.01**	**0.4**	**0.2**	**0.77**	**10^-3^**	0.5	0.23	1.1	0.10	**0.2**	**0.1**	**0.75**	**0.01**	0.65
*HLA-C*	31237124T>C (rs1130838)	**1.4**	**1.1**	**1.77**	**0.02**	**2.9**	**1.6**	**5.3**	**10^−3^**	**2.9**	**1.3**	**6.3**	**0.01**	2.23	0.82	6.05	0.12	**1.4**	**1.1**	**1.79**	**0.02**	**1.6**	**1.2**	**2.3**	**6*10^-3^**	1.1	0.75	1.63	0.69	0.43
	31237162C>G (rs35708511)	**1.3**	**1**	**1.76**	**0.03**	**2.9**	**1.5**	**5.1**	**3*10^-3^**	**2.8**	**1.1**	**6.8**	**0.03**	1.98	0.71	5.5	0.19	**1.3**	**1**	**1.73**	**0.03**	**1.6**	**1.1**	**2.3**	**7*10^-3^**	1.01	0.69	1.49	1	0.17
	31237773T>C (rs1130947)	**1.9**	**1.4**	**2.37**	**<10^−4^**	1.28	0.86	1.92	0.24	1.54	1	3.02	0.06	0.96	0.54	1.17	1	**1.3**	**1**	**1.3**	**0.03**	**1.6**	**1.1**	**2.2**	**0.01**	0.97	0.68	1.38	0.85	0.11
	31237774T>G (rs41540512)	**1.9**	**1.4**	**2.37**	**<10^−4^**	1.28	0.86	1.92	0.24	1.54	1	3.02	0.06	0.96	0.54	1.17	1	**1.3**	**1**	**1.3**	**0.03**	**1.6**	**1.1**	**2.2**	**0.01**	0.97	0.68	1.38	0.85	0.11
	31237779C>T (rs146911342)	**1.5**	**1.1**	**2.1**	**0.01**	**1.5**	**1.1**	**2.2**	**0.02**	**2.2**	**1.3**	**3.8**	**4*10^-3^**	1.32	0.77	2.28	0.34	**1.5**	**1.1**	**2.1**	**0.01**	**2**	**1.3**	**3.3**	**4*10^-3^**	1.14	0.72	1.82	0.63	0.56
	31237858T>C (rs34794906)	**1.4**	**1.1**	**1.9**	**0.02**	**1.7**	**1.2**	**2.3**	**5*10^-3^**	1.45	0.9	2.34	0.12	1.66	0.99	2.8	0.06	**1.4**	**1.1**	**1.91**	**0.01**	**1.5**	**1**	**2.3**	**0.05**	1.31	0.85	2.02	0.23	0.28
	31237987C>T (rs41556321)	**1.5**	**1.1**	**2.09**	**0.02**	**1.8**	**1.2**	**2.6**	**3*10^-3^**	**1.9**	**1.1**	**3.3**	**0.01**	1.42	0.86	2.46	0.27	**1.5**	**1.1**	**2.05**	**0.02**	**1.7**	**1.1**	**2.8**	**0.03**	1.21	0.74	1.97	0.46	0.47
	31238009TT>CG (rs1131014)	**1.3**	**1**	**1.66**	**0.02**	**1.7**	**1.2**	**2.4**	**2*10^-3^**	1.56	0.98	2.47	0.07	1.48	0.9	2.44	0.13	**1.4**	**1.1**	**1.8**	**0.01**	**1.5**	**1.1**	**2.2**	**0.02**	1.16	0.79	1.7	0.49	0.06
	31238027C>A (rs41540117)	**1.5**	**1.1**	**2.04**	**0.02**	**1.8**	**1.2**	**2.6**	**3*10^-3^**	**1.8**	**1.1**	**3.1**	**0.03**	1.54	0.87	2.67	0.16	**1.5**	**1.1**	**2.12**	**0.01**	**1.8**	**1.1**	**2.9**	**0.02**	1.27	0.78	2.06	0.39	0.49
	31238029T>C (rs2308622)	**1.3**	**1**	**1.62**	**0.04**	**1.6**	**1.1**	**2.4**	**0.02**	1.72	1	2.97	0.07	1.07	0.6	1.93	0.88	**1.3**	**1**	**1.66**	**0.02**	**1.6**	**1.1**	**2.2**	**0.01**	1.05	0.74	1.5	0.78	0.36
	31238138C>T (rs1050320)	**1.3**	**1**	**1.77**	**0.04**	**1.8**	**0.1**	**2.5**	**2*10^-3^**	**2.2**	**1.4**	**3.6**	**2*10^-3^**	1.06	0.63	1.78	0.89	1.22	0.96	1.56	0.11	1.4	1	1.97	0.05	0.98	0.69	1.42	1	0.18
	31239449C>G (rs28626310)	**0.6**	**0.5**	**0.82**	**<10^−4^**	**0.5**	**0.4**	**0.7**	**<10^−4^**	**0.5**	**0.3**	**0.7**	**3*10^-3^**	0.8	0.48	1.32	0.43	**0.1**	**0.4**	**0.8**	**10^-3^**	**0.5**	**0.3**	**0.7**	**<10^−4^**	0.87	0.57	1.32	0.52	0.39
*HLA-F*	29694777G>A (rs17875384)	**3.6**	**1.3**	**10.3**	**0.01**	1.88	0.81	4.33	0.14	2.81	0.64	12.28	0.18	**5.1**	**1.1**	**23**	**0.02**	**3.6**	**1.3**	**10.2**	**0.01**	2.9	0.67	12.53	0.19	**4.9**	**1.1**	**21.9**	**0.02**	0.62
*HLA-DQA1*	32609806C>T (rs707952)	**0.7**	**0.5**	**0.99**	**0.05**	0.66	0.4	1.08	0.10	0.73	0.36	1.49	0.44	0.64	0.3	1.38	0.32	**0.6**	**0.4**	**9.43**	**0.02**	0.67	0.38	1.2	0.20	0.59	0.32	1.1	0.10	0.47
	32609952T>C (rs707949)	**0.7**	**0.5**	**0.96**	**0.03**	0.74	0.53	1.04	0.09	0.7	0.44	1.11	0.14	0.67	0.4	1.13	0.14	0.76	0.57	1.02	0.07	0.76	0.51	1.14	0.20	0.71	0.43	1.12	0.15	0.58
	32609969T>G (rs707963)	**0.7**	**0.5**	**0.95**	**0.02**	0.74	0.52	1.03	0.08	0.74	0.46	1.18	0.22	0.62	0.37	1.04	0.08	0.76	0.56	1.01	0.06	0.8	0.53	1.19	0.28	0.67	0.43	1.06	0.10	0.71
	32609974T>G (rs707962)	**0.7**	**0.5**	**0.92**	**0.01**	0.71	0.5	0.99	0.05	0.72	0.45	1.15	0.18	**0.6**	**0.4**	**0.98**	**0.05**	**0.7**	**0.6**	**0.98**	**0.04**	0.78	0.52	1.17	0.23	0.64	0.41	1.01	0.06	0.66
*HLA-DRB1*	32551939GG>AT (rs16822512)	**0.8**	**0.6**	**0.97**	**0.03**	0.83	0.59	1.15	0.27	**0.6**	**0.4**	**1**	**0.05**	0.89	0.54	1.47	0.70	**0.8**	**0.6**	**0.97**	**0.03**	0.72	0.51	1.02	0.66	0.75	0.51	1.11	0.16	0.59
	32552092A>T (rs1059569)	**1.5**	**1.1**	**3.07**	**0.05**	**1.8**	**1.2**	**2.9**	**0.01**	**2.2**	**1.1**	**4.4**	**0.03**	1.17	0.6	2.3	0.73	**1.6**	**1**	**2.46**	**0.04**	1.91	0.99	3.68	0.05	1.3	0.69	2.45	0.43	0.62
*HAL-DRB5*	32487256G>A (rs112401921)	**0.5**	**0.3**	**0.94**	**0.03**	**0.4**	**0.2**	**0.9**	**0.05**	**0.2**	**0.6**	**0.1**	**6*10^-3^**	0.84	0.22	3.21	1	**0.4**	**0.2**	**0.87**	**0.02**	**0.3**	**0.1**	**0.6**	**4*10^-3^**	0.95	0.3	3.03	1	0.40

The two *trans* (i.e., in each copy of the single HLA genes) alleles were included as covariates in a log-additive model. Significant HLA-SNPs were analyzed by genotypic (chi squared testing from 2 × 3 contingency table) and allelic frequencies (2 × 2 table chi squared testing), in the whole population or stratified by biological sex. As for conventional alleles (Table 2), no statistically significant differences were seen in SNP distribution between sexes in the whole population, ruling out possible sampling bias (see the *p*-value in M vs. F column). The table reported odds ratio (OR), lower (Low) and upper (Up) confidence intervals (CI) and *p*-values for each analysis. Bolded SNPs in coloured cells were significant.

**Table 5 ijms-25-13198-t005:** HLA super haplotypes associated with COVID-19 severity in the whole population: males and females.

*HLA* Gene	Haplotype ID #	Whole Population						Males						Females						M vs. F
		# of Alleles		OR	*CI*		p	# of Alleles		OR	*CI*		p	# of Alleles		OR	*CI*		p	p
		Cases *n =* 964	Ctrls *n =* 396		Low	Up		Cases *n =* 654	Ctrls *n =* 186		Low	Up		Cases *n =* 310	Ctrls *n =* 210		Low	Up		
*-B*	3	633	270	0.89	0.7	1.15	0.41	**411**	**144**	**0.49**	**0.34**	**0.72**	**3*10^-4^**	**222**	**126**	**1.68**	**1.16**	**2.44**	**0.01**	0.75
	5	184	75	1.01	0.75	1.36	1	**121**	**48**	**0.65**	**0.44**	**0.96**	**0.04**	**63**	**27**	**1.73**	**1.06**	**2.82**	**0.04**	0.2
	9	**23**	**1**	**9.65**	**1.3**	**71.74**	**0.01**	11	1	3.16	0.41	24.67	0.42	**12**	**0**	**-**	**-**	**-**	**0.01**	0.23
	15	**53**	**11**	**2.04**	**1.05**	**3.94**	**0.04**	35	5	2.05	0.79	5.3	0.19	18	6	2.1	0.82	5.37	0.17	0.9
*-C*	4	**114**	**73**	**0.59**	**0.43**	**0.82**	**2*10^-3^**	**80**	**40**	**0.51**	**0.33**	**0.77**	**2*10^-3^**	34	33	0.66	0.39	1.11	0.15	0.47
	8	66	23	1.19	0.73	1.95	0.56	42	17	0.68	0.38	1.23	0.26	**24**	**6**	**2.85**	**1.15**	**7.11**	**0.03**	0.36
	17	73	23	1.33	0.82	2.16	0.30	46	15	0.86	0.47	1.58	0.75	**27**	**8**	**2.41**	**1.07**	**5.41**	**0.04**	0.71
	19	2	4	0.2	0.04	1.12	0.11	**1**	**4**	**0.07**	**0.01**	**0.63**	**0.01**	1	0	-	-	-	1	0.28
	44	**1**	**4**	**0.1**	**0.01**	**0.91**	**0.04**	1	1	0.28	0.02	4.55	0.92	0	3	-	-	-	0.13	0.32
*-F*	1	**34**	**4**	**3.58**	**1.26**	**10.16**	**0.01**	20	2	2.9	0.67	12.53	0.22	**14**	**2**	**4.92**	**1.11**	**21.87**	**0.04**	0.74
*-DQA1*	1	**784**	**299**	**1.41**	**1.07**	**1.87**	**0.02**	**503**	**158**	**0.59**	**0.38**	**0.92**	**0.02**	**281**	**141**	**4.74**	**2.94**	**7.65**	**4*10^-11^**	0.27
	2	111	56	0.79	0.56	1.12	0.21	**74**	**35**	**0.55**	**0.35**	**0.85**	**0.01**	37	21	1.22	0.69	2.15	0.59	0.32
*-DRB1*	1	665	257	1.2	1.54	0.94	0.14	430	136	0.71	1.01	0.49	0.06	**235**	**121**	**2.3**	**3.36**	**1.58**	**10^-4^**	0.68
	**2**	**22**	**18**	**0.49**	**0.26**	**0.92**	**0.04**	**15**	**11**	**0.37**	**0.17**	**0.83**	**0.02**	7	7	0.67	0.23	1.94	0.64	0.25
	3	**193**	**112**	**0.63**	**0.48**	**0.83**	**10^-3^**	**117**	**62**	**0.44**	**0.3**	**0.63**	**10^-5^**	76	50	1.04	0.69	1.57	0.94	0.21
*-DRB5*	1	**13**	**13**	**0.4**	**0.18**	**0.88**	**0.03**	**6**	**8**	**0.21**	**0.07**	**0.6**	**10^-3^**	7	5	0.95	0.3	3.03	1	0.53

No statistically significant differences were seen in HLA haplotype distribution between sexes in the whole population, ruling out possible sampling bias (see the *p*-value in M vs. F column). The table reports odds ratio (OR), lower (Low) and upper (Up) confidence intervals (CI) and *p*-values for each analysis. Bolded SNPs in coloured cells were significant.

## Data Availability

Data available in the Appendix A.

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
