# Peer review of "Sex-Specific HLA Alleles Contribute to the Modulation of COVID-19 Severity"

_ijms, 2024, doi:10.3390/ijms252313198_

Round 1
Reviewer 1 Report
Comments and Suggestions for Authors
The manuscript “Sex specific HLA alleles contribute to the modulation of Covid- 2 19 severity” investigates the association between Covid-19 severity (severe vs. asympto- 30 matic/oligosymptomatic healed individuals) and HLA gene variants, analyzed by next-generation 31 sequencing.The results have reference value, but there are many problems in the writing and the presentation of the results.
1. The information in the abstract is not specific enough, and detail information about important results should be listed.
2. Some directly related articles were not referenced in the article, such as PMID38251811.
3. Males in cases group is 67.55%, high than 42.47% in control.
4. In line 97-97, the content in this section does not match the sub-title. The results did not mention the impact of gender differences.
5. In line 101-102, there is no result of HLA-F, HLA-DQA 1, HLA-DRB5 in table 2. In “Whole population” column, why marker A and DPA1 lines? In “Females” column, why marker DPB1 line?
6. In line 120-121, why there are two different titles for table 3? What does bold font in Table 3 represent?
7. Did the author analyze the impact of gender on patients with same age?
8. In table 5, “n.210” change to “n=210”.
9. In line 120-121, inappropriate expression for “our data provide a first evi- 254 dence of biological sex specific differences in disease susceptibility related to HLA genes 255 in humans. ”. There have been multiple articles discussing and analyzing this point.
Comments on the Quality of English LanguageLanguage is acceptable. But the writing of the results section needs to be improved.
Author Response
”Sex specific HLA alleles contribute to the modulation of COVID-19 severity”.
Response to Reviewer 1 Comments
|
|||||||||||||||
1. Summary |
|
|
|||||||||||||
Thank you very much for taking the time to review this manuscript. Please find the detailed responses below and the corresponding revisions/corrections highlighted/in track changes in the re-submitted files.
|
|||||||||||||||
2. Questions for General Evaluation |
Reviewer’s Evaluation |
Response and Revisions |
|||||||||||||
Does the introduction provide sufficient background and include all relevant references? |
Can be improved |
Introduction has been extended and some references added |
|||||||||||||
Are all the cited references relevant to the research? |
Yes |
|
|||||||||||||
Is the research design appropriate? |
Yes |
|
|||||||||||||
Are the methods adequately described? |
Can be improved |
Methods have been extended and further detailed |
|||||||||||||
Are the results clearly presented? |
Must be improved |
Results have been extended, pointing on comments on tables |
|||||||||||||
Are the conclusions supported by the results? |
Can be improved
|
The discussion has been deeply revised and rephrased |
|||||||||||||
3. Point-by-point response to Comments and Suggestions for Authors |
|||||||||||||||
Comments 1: The information in the abstract is not specific enough, and detail information about important results should be listed. |
|||||||||||||||
Response 1: following the suggestions of the reviewer, the abstract has been rephrased |
|||||||||||||||
Comments 2: Some directly related articles were not referenced in the article, such as PMID38251811. |
|||||||||||||||
Response 2: Thank you very much for the suggestion. We missed this reference that has been now included in the manuscript.
|
|||||||||||||||
4. Response to Comments on the Quality of English Language |
|||||||||||||||
Point 1: Language is acceptable. But the writing of the results section needs to be improved. |
|||||||||||||||
Response 1: Thank you, we have modified the text accordingly.
|
Reviewer 2 Report
Comments and Suggestions for Authors
Thanks for the opportunity to review the manuscript by Serena Spartano and Cols.
The authors aimed to investigate the association between COVID-19 severity and HLA alleles analyzed by NGS. They identified alleles, SNPs, and haplotypes associated with COVID-19 severity related to biological sexes. Also, they identified specific haplotypes ("super-haplotypes") shared by different HLA alleles that, they argue, have a sex-specific impact on COVID-19 risk.
The research was impressive: methodological, hands-on, and bioinformatics analyses are well-designed and described adequately.
I have some comments that I hope the authors find helpful.
Throughout the entire manuscript, please correct Covid-19 by COVID-19.
Supplemental Figure 1 is really depicting, I recommend include in the main manuscript file.
In line 112, you stated: We prioritized coding and intronic variants that could affect exon splicing: 28 variants. Was this independent of the allele frequency of the selected variants? Please comment on the text and its meaning and importance, if any.
In Table 4, better than gDNA ID, include the "rs" identifier in Table 4.
In different sections, you propose that your study has been conceived as retrospective with prospective recruitment. So, could it be appropriate to mention it as prolective? Please consider it.
In the discussion section, move the limitations paragraph to the end of the section.
A couple of lines explaining the meaning of cis— / trans— variants would be helpful for non-expert readers. Please add it.
In line 303, include the rs ID for the SNP in TBC1D4.
The conclusion paragraph, "In conclusion, our study provides some hints that will likely open new scenarios in understanding HLA-mediated immune modulation in response to environmental triggers, beyond the SARS-CoV2 pandemics." should be:
a) Separated from the rest of the remaining paragraph, and
b) Reformulated, these lines can be part of the conclusion as perspectives but are not your conclusions. This should include the theoretical (biological, clinical, evolutionary, etc.) interpretation of your main results, and maybe later "...provides some hints that will likely open new scenarios in understanding HLA-mediated immune modulation in response to environmental".
Author Response
”Sex specific HLA alleles contribute to the modulation of COVID-19 severity”.
Response to Reviewer 2 Comments
|
|||||||||||||||
1. Summary |
|
|
|||||||||||||
Thank you very much for taking the time to review this manuscript. Please find the detailed responses below and the corresponding revisions/corrections highlighted/in track changes in the re-submitted files.
|
|||||||||||||||
2. Questions for General Evaluation |
Reviewer’s Evaluation |
Response and Revisions |
|||||||||||||
Does the introduction provide sufficient background and include all relevant references? |
Yes |
|
|||||||||||||
Are all the cited references relevant to the research? |
Yes |
|
|||||||||||||
Is the research design appropriate? |
Yes |
|
|||||||||||||
Are the methods adequately described? |
Yes |
Methods have been extended and further detailed |
|||||||||||||
Are the results clearly presented? |
Can be improved |
Results have been extended, pointing on comments on tables |
|||||||||||||
Are the conclusions supported by the results? |
Can be improved
|
The discussion has been deeply revised and rephrased |
|||||||||||||
3. Point-by-point response to Comments and Suggestions for Authors |
|||||||||||||||
Comments 1: Throughout the entire manuscript, please correct COVID-19 by COVID-19. |
|||||||||||||||
Response 1: Thank you, we have modified the text accordingly. |
|||||||||||||||
Comments 2: Supplemental Figure 1 is really depicting, I recommend include in the main manuscript file. |
|||||||||||||||
Response 2: We appreciate the comment of the reviewer. The figure has been moved to the main text.
|
|||||||||||||||
|
Reviewer 3 Report
Comments and Suggestions for Authors
The work presented by Spartano et al is very well structured and interesting. However, I have a few minor comments for the authors.
Major:
1) Line 139: Table3D is missing. Is this a typo and do you mean table 3?
2) I would prefer the whole ‘materials and methods’ section to be written in more detail.
Minor:
1) The text must be organised according to the journal guidelines.
2) Covid-19 should be standardised throughout the text. I would ask the authors to write it in all caps (COVID-19).
3) Line 38: write ‘immuno-responses’, the hyphen improves reading.
4) Lines 49-50: write SARS-CoV-2, not write SARS-CoV2.
5) All tables appear with two descriptions. Please standardise according to the style of the paper.
6) Line 117: OR and IC acronyms appear without first describing what they are. They also appear in the tables. These acronyms, however well known, should also be explained in the notes of the tables.
7) Line 145: the word ‘haplotypes’ already appears previously. Fix the position of the acronym.
8) Lines 165-166 and 181: extra spaces.
9) Lines 254-256: writing in an article that it is the first evidence is not correct even though it actually is. Please reword the sentence.
10) Line 344: a space is missing between ‘>2.The’.
11) Line 388: the acronym OR has already been used (see comment 6 minor).
Extra:
1) Lines 187-197: The topic is very interesting. I look forward to future insights from you.
Author Response
”Sex specific HLA alleles contribute to the modulation of COVID-19 severity”.
Response to Reviewer 3 Comments
|
|||||||||||||||
1. Summary |
|
|
|||||||||||||
Thank you very much for taking the time to review this manuscript. Please find the detailed responses below and the corresponding revisions/corrections highlighted/in track changes in the re-submitted files.
|
|||||||||||||||
2. Questions for General Evaluation |
Reviewer’s Evaluation |
Response and Revisions |
|||||||||||||
Does the introduction provide sufficient background and include all relevant references? |
Yes |
|
|||||||||||||
Are all the cited references relevant to the research? |
Yes |
|
|||||||||||||
Is the research design appropriate? |
Yes |
|
|||||||||||||
Are the methods adequately described? |
Must be improved |
Methods have been extended and further detailed |
|||||||||||||
Are the results clearly presented? |
Yes |
Results have been extended, pointing on comments on tables |
|||||||||||||
Are the conclusions supported by the results? |
Yes
|
The discussion has been deeply revised and rephrased |
|||||||||||||
3. Point-by-point response to Comments and Suggestions for Authors MAJOR: |
|||||||||||||||
Comments 1: Line 139: Table3D is missing. Is this a typo and do you mean table 3? |
|||||||||||||||
Response 1: We apologize for the typo. Table 2 was inappropriately indicated as Table 3D
|
|||||||||||||||
Comments 2: I would prefer the whole ‘materials and methods’ section to be written in more detail. |
|||||||||||||||
The text was extended and further detailed as requested.
MINOR:
|
|||||||||||||||
|
Round 2
Reviewer 2 Report
Comments and Suggestions for Authors
Thanks for attending to my previous concerns.
Reviewer 3 Report
Comments and Suggestions for Authors
The article is greatly improved and can be accepted.